# Circular RNAs, Noncoding RNAs, and N6-methyladenosine Involved in the Development of MAFLD

**DOI:** 10.3390/ncrna10010011

**Published:** 2024-02-05

**Authors:** Moeka Nakashima, Naoko Suga, Yuka Ikeda, Sayuri Yoshikawa, Satoru Matsuda

**Affiliations:** Department of Food Science and Nutrition, Nara Women’s University, Kita-Uoya Nishimachi, Nara 630-8506, Japan

**Keywords:** N6-methyladenosine, noncoding RNA, circular RNA, RNA-binding protein, APRO family protein, NAFLD, MAFLD, probiotics, YTHDF2

## Abstract

Noncoding RNAs (ncRNAs), including circular RNAs (circRNAs) and N6-methyladenosine (m6A), have been shown to play a critical role in the development of various diseases including obesity and metabolic disorder-associated fatty liver disease (MAFLD). Obesity is a chronic disease caused by excessive fat accumulation in the body, which has recently become more prevalent and is the foremost risk factor for MAFLD. Causes of obesity may involve the interaction of genetic, behavioral, and social factors. m6A RNA methylation might add a novel inspiration for understanding the development of obesity and MAFLD with post-transcriptional regulation of gene expression. In particular, circRNAs, microRNAs (miRNAs), and m6A might be implicated in the progression of MAFLD. Interestingly, m6A modification can modulate the translation, degradation, and other functions of ncRNAs. miRNAs/circRNAs can also modulate m6A modifications by affecting writers, erasers, and readers. In turn, ncRNAs could modulate the expression of m6A regulators in different ways. However, there is limited evidence on how these ncRNAs and m6A interact to affect the promotion of liver diseases. It seems that m6A can occur in DNA, RNA, and proteins that may be associated with several biological properties. This study provides a mechanistic understanding of the association of m6A modification and ncRNAs with liver diseases, especially for MAFLD. Comprehension of the association between m6A modification and ncRNAs may contribute to the development of treatment tactics for MAFLD.

## 1. Introduction

Circular RNAs (circRNAs) are a class of single-strand closed RNA molecules [1]. They are generated by the reverse splicing process, which may be involved in the development of many diseases, including various liver diseases. Most circRNAs are non-coding RNAs (ncRNAs), while part of cytoplasmic circRNAs with coding capability can be translated into peptides, which may contribute to several important biological and/or physiological processes since many circRNAs originate from exons and reside in the cytoplasm [2]. Therefore, innovative research for circRNA-based therapeutic and/or diagnostic strategies has been conducted [3]. circRNAs are also observed in eukaryotic cells [4], whose expression pattern might enable them to support the clinical diagnosis as biological markers in many types of disease [5]. In particular, circRNAs are highly stable and easily detected in the circulation, which has been shown to be valuable as non-coding RNAs represent a promising non-invasive approach for predicting a non-alcoholic fatty liver disease (NAFLD) [6].

Generally, ncRNAs may lack the capability of translating into protein/peptide; however, ncRNAs can work as crucial transcripts to regulate the expression of other genes [7]. These ncRNAs may be typically divided into several groups, including microRNAs (miRNAs), long ncRNAs (lncRNAs), and circRNAs [8]. In addition, the n6-methyladenosine (m6A) modification in those ncRNAs has been increasingly described to have a dense association with the pathological machinery of various diseases [9]. Several crucial regulators derived from m6A modification could affect the profound function of ncRNAs to take part in the initiation and/or progression of related diseases. Remarkably, ncRNAs can mediate downstream signaling to impact the expression of readers, writers, and/or erasers for the additional modification of m6A functions [9]. m6A RNA modification may highly regulate hepatic function and the development of liver diseases, providing some directions to understand the mechanism of malicious activities in the development of liver diseases [10] (Figure 1).

m6A, the most prevalent modification associated with eukaryotic RNAs, can affect several steps of RNA metabolism, including translation, splicing, and the decay/stability of RNAs, which may be the epitranscriptomic modification that occurs on the N6 position of adenosine [11]. It is the most abundant modification in eukaryotic RNAs, accounting for about 0.4% of all adenosines in RNAs, which may be involved in the functional alteration of RNAs [12]. Interestingly, m6A modification may be closely associated with the initiation and/or development of obesity and non-alcoholic fatty liver disease (NAFLD), which may progress to end-stage liver disease [13]. Obesity is the main risk factor for NAFLD. The feature of the disease is hepatic steatosis with the accumulation of surplus fat in the liver and metabolic liver dysfunction. Therefore, it has been suggested that NAFLD should be retitled as metabolic disorder-associated fatty liver disease (MAFLD). Here, we use the term MAFLD instead of NAFLD. MAFLD is often assumed to be asymptomatic. However, many MAFLD patients complain of exhaustion, which may disturb their quality of life (QOL). Currently, there are no specific pharmacotherapies for MAFLD. Mostly, the treatment may include lifestyle adjustments and medicines for improving fat metabolism and balancing oxidation. Therefore, further research and development of novel therapeutic tactics are required [13]. In addition, obesity is a chronic metabolic disease that is closely related to type 2 diabetes mellitus, cardiovascular diseases, and osteoarthritis [14]. The prevalence of obesity is increasing rapidly, which is considered to be a global public health burden [14]. However, the precise mechanism and role of m6A-modified ncRNAs in MAFLD are not well understood. m6A modification may play a role in the formation of specific and complexed microenvironments in the liver [15]. In fact, recent studies have shown that nucleotide methylation may be directly associated with the inflammatory grade, lipid synthesis, and/or oxidative stress, playing a crucial role in the progression of MAFLD [15]. Therefore, m6A modification may be a key player in the pathogenesis of MAFLD, which may provide new mechanistic insight into MAFLD [16]. Recent studies have explored the roles of m6A RNA methylation in the pathogenesis of liver diseases, providing new insights for studying the molecular mechanism of liver diseases [17]. Epigenetic modification in RNA has become the hotspot of the field, offering the potential of m6A as a treatment option for several liver diseases.

## 2. Association of m6A and ncRNAs with Various Liver Diseases

M6A may be one of the highly prevalent modifications in various mRNAs and/or ncRNAs that affect the epigenetic effects, which could play a critical role in cellular physiology and/or pathology. The effects of m6A are determined by reader (recognition), writer (methyltransferase), and eraser (demethylase) molecules [18] (Figure 1). Examples of writers are special methyltransferases, including Wilms tumor 1 associated protein (WTAP), methyltransferase-like 3 (METTL3), Vir-like m6A methyltransferase-associated (VIRMA), and methyltransferase-like 14 (METTL14). The AlkB homolog 5 (ALKBH5) and fat mass and obesity-associated protein (FTO) are known examples of erasers, which are special demethylases that can reverse m6A methylation. m6A modifications are regularly recognized by m6A-binding reader molecules, including YT521-B homology (YTH) domain-containing-1 and -2 (YTHDC-1 and -2) and YTH domain family proteins 1-3 (YTHDF1-3). In addition, the heterogeneous nuclear ribonucleoprotein (HNRNP) family, including HNRNPC and heterogeneous nuclear ribonucleoprotein A2/B1 (HNRNPA2B1), may have also a YTH domain for binding the m6A site of m6A modified RNAs [19]. It has been suggested that eukaryotic initiation factor 3 (eIF3) may also initiate the translation in a cap-independent manner by binding to the m6A sites in the 5′-UTR of mRNAs, while insulin-like growth factor 2 mRNA-binding protein 1/2/3 (IGF2BP1/2/3) can increase the stability of the target RNAs [19]. m6A modification may play an essential role in the proliferation and/or differentiation of hepatocytes, which are essential to the liver response to injury and regeneration. For example, m6A content might be decreased in patients with type 2 diabetes, and mRNA expression levels of FTO, METTL3, METTL14, and WTAP seem to be enhanced [20]. Remarkably, m6A content may be negatively associated with the expression level of METTL3, METTL14, and FTO mRNA, which could play an important role in the proliferation and/or differentiation of hepatocytes. High glucose can also upregulate the protein expression of FTO in HepG2 cells [20]. In addition, the FTO could decrease the concentration of m6A in RNA transcripts, thereby regulating the expression of related transcripts [21]. It has been demonstrated that m6A modification has an impact on the regulation of carboxylesterase 2 (CES2), a serine esterase responsible for the hydrolysis of endogenous substrates, including triglycerides and diacylglycerides, affecting lipid metabolism [22]. Furthermore, the phosphorylation of p70 ribosomal subunit 6 kinase (S6K) and/or mammalian target of rapamycin (mTOR) may be reduced by the knockout or knockdown of WTAP [23].

Circular RNAs (circRNAs) can also play a part in physiological and/or pathological processes through a similar pathway as a new type of molecule with intriguing functions [24], which may represent a favorable non-invasive approach for predicting MAFLD [6]. In addition, proteins encoded by circRNAs have been confirmed to be related to multiple pathophysiological processes, including immunity [25]. m6A modifications can regulate the metabolism of circRNAs. It has been shown that circRNAs containing m6A residues can be translated into protein by non-cap-dependent structures [26]. m6A-modified circRNA might be recognized by the YTHDF2 [27], in which the open reading framework (ORF) may be verified for the translational capability of the circRNA. Several effects of the peptide/protein from circRNA translation have been identified [28]. FTO demethylase could diminish the rate of circRNA translation, but METTL3 methyltransferase can boost the translation rate [29].

Remarkably, these proteins may also be significant players in the development of MAFLD. In particular, m6A methyltransferase is known to exert regulatory functions in liver-related diseases [30]. The liver is a vital metabolic and digestive organ in the pathophysiological processes. Recent studies have suggested that m6A RNA modification can regulate hepatic function and the development of liver diseases [10,31]. For example, METTL14 can modulate the expression of miR-34a-5p, impairing mitochondrial homeostasis in MAFLD [31]. In addition, many of these epigenetic factors are amenable to dietary or lifestyle interventions. For example, several interventions may include diets low in carbohydrates, free sugars, fructose, and lipids, in addition to healthy eating patterns and probiotics [32]. At diverse levels, the nutritional and metabolic status of individuals may interact with epigenetics, which are chemical modifications that affect gene expression without altering DNA sequences [33]. Therefore, it is important to understand how we can control epigenetic characteristics through both lifestyle habits and some clinical interventions. Misbalanced epigenetic regulation can result in various metabolic diseases and/or aging [33]. Epigenetic modifications are potentially reversible and can provide new therapeutic avenues for treating these diseases using epigenetic modulators [34]. Many studies have shown that m6A modification is closely related to the pathology of MAFLD [35]. Recently, studies have found that the regulation of immune cells by m6A modification may also play a role in MAFLD.

## 3. ncRNAs and RNA-Binding Proteins Involved in mRNA Translation

Accumulating evidence has shown that circRNAs act as a sponge to absorb miRNA function, pointing to their connection in the biological process of MAFLD [36,37]. In fact, several circRNAs have been found to be correlated with miRNAs for mediating the expression of m6A regulators. In addition, circRNAs may adjust cellular function by multiple pathways, which may compete against the target endogenous RNAs such as miRNAs, known as sponging [38]. For example, it has been shown that Circ_0057558 can act as a miR-206 sponge to inhibit the repression of the ROCK1/AMPK signaling and facilitate lipogenesis, which may contribute to the progression of MAFLD [37]. A single circRNA could attach to one or more miRNAs through its circular sequences. The sponging of miRNA could play a key role within several types of cells in various diseases. Therefore, translational control of RNAs may be a critical factor for the related disease [39].

In the translation of mRNAs, the circularization by binding eIF4G on the 5′-cap of the mRNA and poly(A) binding protein (PABP) on the poly(A) tail may start the mRNA translation. Therefore, obstructing the interaction between eIF4G and PABP might constrain the mRNA translation. In addition, previous studies have shown that PABP-interacting protein 2 (Paip2) can take part in the eIF4G binding to bind to the PABP with a common amino acid sequence of PABP [40]. Then, several miRNAs could be associated with AGO proteins within the complementary 3′ UTR sites. The AGO-miRNA complex is additionally modified by adjacent RNA-binding proteins (RBPs). circRNAs can also control the gene expression of RNAs by attaching targeted miRNAs in several cells including immune cells and/or cancer stem cells [41]. Therefore, several circRNAs can contribute to the development of MAFLD [36,37]. In some cases, certain circRNA may also impede the translation by binding to neighboring PABP [42]. The association between circRNAs and PABP may impact the combination of PABP and eIF4G, which may specifically influence the translation of several mRNAs [42] (Figure 2).

Several members of the APRO family have been shown to be implicated in cytoplasmic mRNA deadenylation and/or RNA turnover [43,44]. Interestingly, the association between APRO family proteins and PABP has also been discovered [45,46]. The N-terminal conserved APRO domain can bind to DNA-binding transcription factors as well as deadenylase subunits of the CCR4/NOT complex [47]. Likewise, some of the APRO family proteins can interact with the cytoplasmic PABP and the poly(A) nuclease complex CCR4/CAF1 [45,46]. APRO proteins, such as Tob1 and Tob2, can concurrently interact with CAF1 protein and PABP, which may trigger the deadenylation of several mRNAs [46]. In addition, Tob1 and Tob2 proteins have a long C-terminal domain with two PAM2 motifs [48]. The antiproliferative effects of Tob1 have been proposed to be involved in the utilization of the CAF1/CCR4 complex [49], which may indicate that APRO proteins exert their function by modulating the metabolism and/or turnover of mRNAs [50]. Another member of the APRO family, BTG2, can also interact with CAF1 deadenylase through its APRO domain to regulate cell proliferation [51]. Similarly, the destabilization of mRNAs by BTG1 and BTG2 may contribute to cell quiescence [52]. circRNAs and miRNAs may inhibit the expression of various mRNAs related to MAFLD, which may employ the mRNAs destabilization with APRO family proteins [53]. Therefore, the miRISC complex can interact with PABP, CAF1, and/or CCR4 deadenylases [54]. Importantly, an important component of the miRISC might be the PABP and APRO family proteins, which may be necessary for miRNA-mediated deadenylation [54]. Since APRO family proteins have the potential to interact with the CCR4/CAF1 complex, they can also be a significant modulator of miRNAs and circRNAs. Remarkably, the expression of APRO family molecules can also be regulated by certain miRNAs and/or circRNAs [55] (Figure 2).

Interestingly, it has been reported that the overexpression of miR-183-5p could lead to upregulation of several lipogenic genes for hepatic triglyceride (TG) accumulation in mice, whereas the inhibition of miR-183-5p may improve or reduce hepatic TG accumulation [56,57]. In this case, Btg1, a member of the APRO family, may be a direct target of miR-183-5p [56]. In addition, BTG2 signaling could play an important role in the development of MAFLD via the regulation of endoplasmic reticulum stress and inflammation. Moreover, NF-κB pathway activation can upregulate BTG2 expression under stress conditions [58]. The correlation between circRNA-HIPK3 and miRNA-29a as well as the correlation between circRNA-0046367 and miRNA-34a expression can influence the regulation of the Wnt/β catenin pathway, which may provide a new target for the treatment of MAFLD [59], in which circRNA sponges may be provided with miRNA-binding sites; thus, the overexpression of circRNA can lead to miRNA suppression and/or upregulation of miRNA target gene expression [60]. In these ways, several studies have reported the role of circRNAs in reducing lipotoxicity and oxidative stress in MAFLD and/or NASH.

## 4. YTHDF2, an m6A Reader Molecule, in the Pathogenesis of Obesity and MAFLD

It is probable that CCR4/NOT-mediated deadenylation and subsequent 3′-to-5′ exoribonucleolytic degeneration of RNAs can also start the degradation of m6A mRNAs. However, m6A-containing mRNAs might undergo several different pathways of RNA degradation, which may involve deadenylation via YTHDF2, an m6A reader protein-CCR4/NOT (deadenylase) complex as well as via the endoribonucleolytic cleavage by the YTHDF2-ribonuclease (RNase) complex. Consequently, another pathway may be the endoribonucleolytic cleavage by the YTHDF2-RNase complex [61]. Suppressed levels of readers/erasers such as YTHDF2 may lead to a hypo-metabolic state in obesity [62]. It has been described that YTHDF2 could regulate specific mRNA degradation. m6A modification may contribute to metabolic reprogramming in MAFLD, which may play a major role in regulating the innate immune responses [63]. It has been shown that m6A modification is critical in the control of macrophage-directed metabolic programming through the regulation of immune transcripts in MAFLD and obesity [63]. *YTHDF2* can bind transcripts carrying m6A in 3′UTR to induce mRNA degradation partially by recruiting the CCR4-NOT deadenylase complex [64]. The binding site of YTHDF2 on m6A may be usually positioned in the 3′-UTR of mRNAs [64,65]. If m6A is located at the 5′-UTR of mRNAs, YTHDF2 can subsequently facilitate protein translation [66]. Interestingly, it has been suggested that resveratrol may alleviate liver lipid metabolism disorders through the changes of m6A levels in the liver of mice [67]. Moreover, changes in m6A RNA methylation and lipid metabolism in various tissues may be interrelated mechanisms [68]. Again, YTHDF2 could modulate the degradation of several mRNAs containing m6A [69]. Remarkably, YTHDF2 can competitively interact with YTHDF3 to accelerate the degradation of specific mRNAs through Argonaute 2 (Ago2) in a manner independent of m6A [70]. Ago2 is the chief component of the RNA-induced silencing complex. It has been shown that hepatic Ago2 regulates the function of peroxisome proliferator-activated receptor α (PPARα) for oxidative metabolism [71]. YTHDF2 can also control glucose metabolism in the development of various diseases [72] (Figure 2).

Some m6A-containing circular RNAs could associate with YTHDF2, which may be downregulated by certain RNases [73]. YTHDF2 can directly interact with the SH domain of CNOT1 to recruit the CCR4-NOT complex, thus inducing the deadenylation and/or degradation of m6A-modified circRNAs [74]. m6A modification and circRNAs could play a key role in the progression of various diseases, including MAFLD, which may contribute to some clinical features. For example, m6A methylation and circRNAs can be employed as novel biomarkers for diagnosis and/or therapy prognosis [75]. Several circRNAs could function as sponges of miRNAs, which may play a major role in the pathogenesis of MAFLD by affecting fundamental genes such as *PPARs* [76]. More studies are required to further explore the relationship between circRNAs and m6A modification in the development of MAFLD.

## 5. Relationship between YTHDF2 and Diabetes Mellitus, Obesity or MAFLD

Diabetes mellitus (DM) is a systemic metabolic disease, where the body becomes incapable of generating an appropriate quantity of insulin. DM also carries a clinical burden as it is associated with obesity, metabolic syndrome, and MAFLD. There is also an enormous economic burden due to healthcare use. Increasing evidence shows that m6A-binding proteins have a vital effect on the development and progression of diabetes [77]. In addition, YTHDF2 and METTL14 may promote hepatic insulin resistance [77]. YTHDC1 can also cause glucose intolerance by regulating the splicing of polyadenylation-specific factor 6 (CPSF6) mRNA [78]. As YTHDF2 is an m6A reading protein that could modulate RNA stability, transcription, and translation, m6A modifications of RNAs recognized with the YTHDF2 might be profoundly implicated in DM progression. For example, activated tumor necrosis factor (TNF) signaling in a tumor microenvironment (TME) may promote YTHDF2 expression, which consequently regulates NF-κB signaling to maintain intra-tumoral Treg survival by accelerating the degradation of m6A-modified transcripts that encode NF-κB-negative regulators [79]. It is well-known that NF-κB signaling is deeply involved in the pathogenesis of DM and/or MAFLD [80]. Therefore, these effects may present new insights into the effect of m6A modification for the treatment of diabetes. Consequently, RNA modifications with N6-methyladenosine m6A have received much attention due to their involvement in the onset and progression of DM, obesity, and/or MAFLD.

It has been reported that miRNA is also a significant bioactive molecule that induces posttranscriptional gene regulation in eukaryotes [81], which can modulate the levels of m6A by targeting the 3′-untranslated region (3′-UTR) of YTHDF2 mRNA in hepatic cells [82]. Interestingly, miR-6125 can inhibit the proliferative ability of colorectal cancer cells by targeting the YTHDF2 mRNA [83], which is also expressed in the cytoplasm of hepatocytes in patients with autoimmune hepatitis [84]. Autoimmune liver diseases, including autoimmune hepatitis as well as MAFLD, may result in an increased risk of DM due to glucose disturbances like insulin resistance [85]. In addition, YTHDF2 can increase the degradation of oncogenes or tumor suppressor genes in a m6A-dependent manner [86]. p53 is well known for its tumor suppressor role, which is associated with an increased risk of DM, obesity, metabolism, and inflammation [87,88]. YTHDF2 and m6A modification could regulate the process of autophagy [89], which may be connected to the development of DM and MAFLD [90]. Interestingly, hepatic gluconeogenesis in the liver might be regulated via the PI3K/AKT signaling pathway [90]. Downregulation of YTHDF2 can enhance the expression of SOCS2, which may negatively regulate the cell proliferation pathway of JAK/STAT signaling [91]. Therefore, YTHDF2 can suppress the phosphorylation of STAT5 for the inhibition of cell growth [92]. Interestingly, miR-223 has been reported to be abnormally expressed in DM, which can regulate inflammation by targeting different targets [93]. In addition, YTHDF2 can enhance the protein expression of OCT4 by retaining m6A methylation of the 5′-UTR of OCT4 mRNA [94]. Pancreatic β cell failure is a hallmark of diabetes, in which OCT4 signaling may be involved [95]. Consequently, YTHDF2 could function by regulating the expression of target genes to influence the development of various diseases, including DM, obesity, and MAFLD. YTHDF2 could also act as a target of miRNA and circRNA to participate in the progression of several diseases. However, the connotation between YTHDF2 and ncRNAs in DM needs further investigation.

## 6. Future Perspectives

Gut microbiota collaborations may play significant roles in the development of MAFLD [96]. In addition, it has been suggested that the gut microbiota possesses a robust impact on the host m6A epitranscriptome. For example, *Fusobacterium nucleatum* in gut microbiota may induce a remarkable decline of m6A modifications by the downregulation of an m6A methyltransferase METTL3 [97]. In addition, the *Fusobacterium nucleatum* can reduce METTL3 transcription, which may promote the expression of a kinesin family member by reducing its m6A levels and diminishing YTHDF2-dependent mRNA degradation [97]. Non-coding RNAs, including circRNAs, have been suggested to be potentially good biomarkers in gut microbiota-associated diseases [98]. Therefore, microbiome engineering as a strategy for improving human health has been deliberated. For example, fecal microbiota transplantation (FMT) from *Bifidobacterium* into mice could significantly suppress the lung metastasis of cancer cells, in which the gut microbiota may mechanistically impact circRNA expression to regulate the levels of resultant miRNAs [99]. Consequently, the gut microbiota may serve as a therapeutic target.

The gut microbiota has also emerged as a key regulator in the pathogenesis of MAFLD for the development of specific therapies [100] (Figure 3). In the presence of microbial dysbiosis, bacteria and their metabolic products may initiate inflammatory pathways, which result in hepatocellular inflammation [101]. There may be a mutual relationship between microbiota and host, which may regulate gene expression mostly due to the production of metabolites that are generated after colonic fermentation of bioactive compounds such as short-chain fatty acids (SCFA) and/or polyphenols [102]. Interestingly, oxyberberine, an oxidative metabolite of berberine mediated by gut microbiota, has been known to be effective against MAFLD via the alteration of PI3K/AKT signaling [103]. This concept might in turn play an important role in the prevention/treatment of various diseases. In fact, the gut microbiota from the use of probiotics and/or FMT can modulate the host response to various therapeutics including novel immunotherapies [104]. Moreover, the interplay between liver and gut microbiota may affect the efficacy of anti-hepatoma treatments in some cases [105]. On the other hand, the dysbiosis of gut microbiota could probably occur due to the use of broad-spectrum antibiotics, which may unfortunately disturb the valuable microenvironment of the gut [106].

Regulators of m6A have not only been verified as a promising treatment but have also been implemented into clinical and actual therapies. The application of potent inhibitors or activators to regulate m6A organizers may disturb the development of obesity from DM, which is beneficial for MAFLD patients. Furthermore, m6A modification and circRNAs could also influence the pathogenesis by affecting the host immune system of patients. Innate immunity is a natural immune defense function during development, with a pivotal role in inflammatory responses [107]. The modification of the m6A gene could affect the role of circRNAs in innate immune regulation. In addition, exogenous circRNAs can precisely trigger an intracellular signaling pathway in immune cells and promote immune activation [108]. Interestingly, non-coding RNAs could exert an impact on the expression of regulator factors of m6A by indirectly modulating downstream targets or directly regulating particular factors of m6A. Modification of immune cells would form a valuable feedback loop to mediate the modified beneficial prognosis. Further studies on the interaction of m6A and circRNAs/ncRNAs could uncover the comprehensive mechanism of MAFLD (Table 1).

## 7. Conclusions

ncRNAs and m6A modification have been shown to play a critical role in the development and/or progression of various liver diseases including MAFLD with the support of miRNA/circRNA and m6A-reader molecules at the post-transcriptional level. Microbiota collaborations may play critical roles in disease progression via the modification of m6A levels within various RNAs, including ncRNAs.

## Figures and Tables

**Figure 1 ncrna-10-00011-f001:**
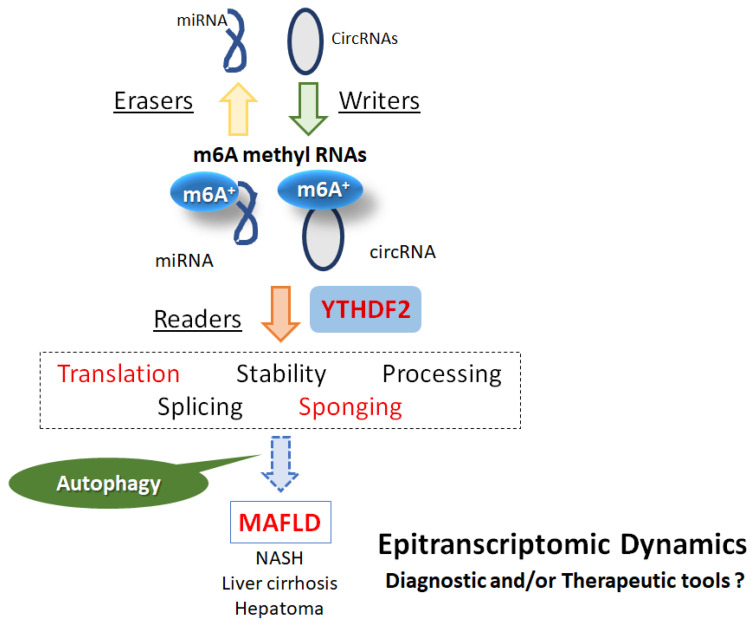
Schematic diagram for the association of noncoding RNAs and m6A methyl RNAs (m6A+) with biological function and liver diseases. In the beginning, m6A modification may be controlled by methyltransferase writers and demethylase erasers. m6A+ binding proteins, including YT521-B homology (YTH) domain-containing-2 (YTHDF2), are called readers, which have been suggested in several RNA activities including translation as well as sponging for the plausible alteration of autophagy. Consequently, m6A methylation in noncoding RNAs could be an important tool for diagnostics and/or therapeutics in several diseases including MAFLD.

**Figure 2 ncrna-10-00011-f002:**
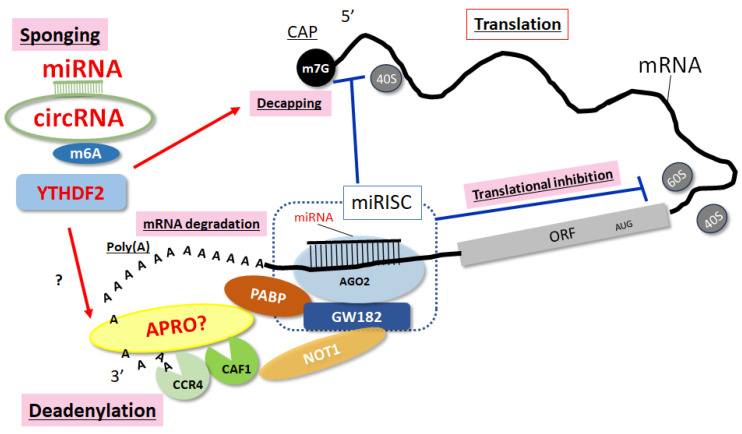
Image of miRNAs, circRNAs, and m6A-mediated inhibition of mRNA translation. RNA-induced silencing complex (miRISC) with certain miRNAs could assist the deadenylation and/or mRNA degradation process by CAF1/CCR4/NOT1 molecules with PABP and APRO protein complexes, in which GW182 and AGO2 might further assemble the miRISK complex. Subsequently, the miRNAs, circRNAs, and m6A with YTHDF2 might play a dynamic role in the regulation of post-transcriptional gene expression via the regulation of mRNA translation and degradation of mRNAs. In attendance, APRO family molecules may interact with PABP to recruit the CAF1/CCR4/NOT1 complex. Red arrowheads indicate stimulation whereas blue hammerheads show inhibition. Note that some critical pathways have been misplaced for clarity. ORF refers to open reading frame and “?” represents author speculation.

**Figure 3 ncrna-10-00011-f003:**
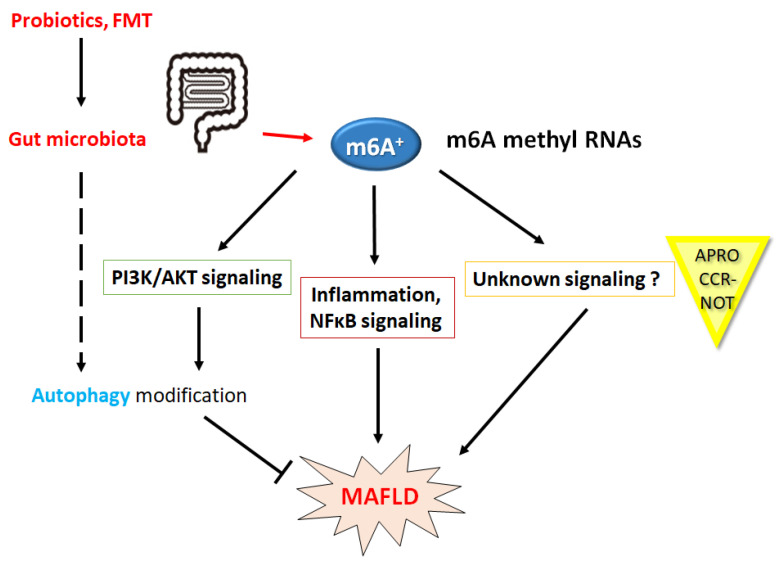
Schematic demonstration of the potential strategies against the pathogenesis of various diseases, including MAFLD. Some kinds of probiotics and/or fecal microbiota transplantation (FMT) might assist the alteration of the gut microbial population for the modification of m6A levels and/or ncRNAs, which might be advantageous for the treatment of several diseases, including MAFLD. Note that some of the important activities such as autophagy initiation, inflammatory reaction, and ROS production have been misplaced for clarity.

**Table 1 ncrna-10-00011-t001:** Some examples of circRNA and/or ncRNA possibly involved in the development of MAFLD.

ncRNA	Expression Pattern in MAFLD	Reference
circ_0057558	increased	[37]
circRNA_0046367	reduced	[59]
miR-34a-5p	increased	[31,59]
miR-183-5p	increased	[56]
miR-206	reduced	[37]
miR-223	reduced	[93]

## Data Availability

Data are contained within the article.

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
