# Peer review of "Circular RNAs, Noncoding RNAs, and N6-methyladenosine Involved in the Development of MAFLD"

_ncrna, 2024, doi:10.3390/ncrna10010011_

Round 1

Reviewer 1 Report

Comments and Suggestions for Authors

The authors in this work use provide an extensive review of literature on the non-coding RNAs like CirRNA with a perspective of RNA modifications like methylation in liver related disorders.

-       Regarding writers and erasers in section 2, lines 96-120 needs to be clear and corrected for content.

-       Same comment as above for lines 213-216.

-       Fig.3 describes PI3K/AKT signaling axis but the text does not have any descriptions.

Minor revisions.

Comments on the Quality of English Language

The authors in this work use provide an extensive review of literature on the non-coding RNAs like CirRNA with a perspective of RNA modifications like methylation in liver related disorders.

-       Regarding writers and erasers in section 2, lines 96-120 needs to be clear and corrected for content.

-       Same comment as above for lines 213-216.

-       Fig.3 describes PI3K/AKT signaling axis but the text does not have any descriptions.

Minor revisions.

Author Response

For Reviewer 1

Comments and Suggestions for Authors

The authors in this work use provide an extensive review of literature on the non-coding RNAs like CirRNA with a perspective of RNA modifications like methylation in liver related disorders.

Thank you so much for the good evaluation to the manuscript.

-       Regarding writers and erasers in section 2, lines 96-120 needs to be clear and corrected for content.

According to this suggestion, the description has been improved. Thank you so much.

-       Same comment as above for lines 213-216.

Accordingly, the description has been improved.

-       Fig.3 describes PI3K/AKT signaling axis but the text does not have any descriptions.

 The PI3K/AKT signaling axis has been described in the section 5 and 6, accordingly.

Minor revisions.

Reviewer 2 Report

Comments and Suggestions for Authors

This manuscript aims to discuss the role of non-coding RNAs and RNA m6A modification in the development of metabolic disorder-associated fatty liver disease (MALFD). The authors introduced the importance of understanding MAFLD biology, the functions and molecular modulators of m6A modification, which are suitable contents for the topic of discussion. However, I found the overall organization of the manuscript needs improvement, as well as its clarity and depth.

1. The title indicates that noncoding RNAs will be the key subject of the review. The authors then started the introduction from circular RNAs (circRNAs), a specific subgroups of RNAs that have both non-coding and coding activities. This could confuse the reader by leaving an impression that circRNAs is the main subject of the review and contradicting the title. According to the title, readers would expect to learn about the involvement of various types of non-coding RNAs in MAFLD. If the authors are only focusing on circRNAs, they should change the title accordingly.

2. The authors tend to discuss the relationship between ncRNA/m6A and MAFLD in a general manner without being specific enough. The authors should explicitly say which circular RNA species and which m6A-modified RNA species impact MAFLD pathology in which direction, through which mechanisms. The authors should avoid statements such as line 223-226 and 260-267. Several good examples are: line 120-123, m6A of CES2 in lipid metabolism; line 142-143, miR-34a-5p in mitochondrial homeostasis (here should introduce the molecular function of miR-34-5p in more detail); line 159-161, circ_0057558 as miR-206 sponge and relevant signaling pathway; line 213-215 miR-183-5p in hepatic triglyceride accumulation, etc. And the authors should organize these pieces of evidences in a logical manner.

3. The authors tend to discuss m6A/ncRNA without in the MAFLD context. For example, line 127-138, line 166-178 and line 189-212 all had this issue. It is necessary to introduce the biological background of the subject, such as the role of circRNA and m6A and how they are processed, but to a brief extent. The key is to discuss the involvement of these processes in MAFLD progression, pathology, prognosis etc with specific examples and mechanistic details.

4. The authors tend to confuse between "m6A in MAFLD" versus "the proteins that read/write/erase m6A in MAFLD". The authors spent two sections discussing YTHDF2, an m6A reader, and MAFLD and Diabetes mellitus. Although YTHDF2 is m6A reader, "m6A in MAFLD" and "YTHDF2 in MAFLD" are two distinctive topics. In "m6A in MAFLD", the subject of discussion should be the RNAs with m6A modification, their biological functions and involvement in MAFLD pathology. The authors should change their title and highlight YTHDF2 in their title if this protein is their subject of discussion, and reorganize the manuscript accordingly. 

5. The content of future perspectives is not suitable. Future perspectives should mainly provide the readers with a summary of the future of the research subject discussed in the manuscript, such as challenges and how to address them, opportunities and how to explore them, etc. The authors discusses microbiota in MAFLD, which is only one aspect of topic, and it lacks the depth required for a future perspective section.

Comments on the Quality of English Language

The English language is fine, except that the authors tend to use too many "might" "could" in the manuscript, such as: might: line 31, 38, 133 and 134, line 167, line 308, etc; could: line 34, 44, 49, 113, 118, 119 etc. When discussing scientific facts that are supported by previous research data, the authors should avoid being ambiguous or too conservative by using "might" and "could", but be explicit about the facts.

There are also some expressions that are not commonly used: line 11, considered by; line 11, has been increased lately; line 20, has been appeared to occur; etc.

Author Response

For Reviewer 2

This manuscript aims to discuss the role of non-coding RNAs and RNA m6A modification in the development of metabolic disorder-associated fatty liver disease (MALFD). The authors introduced the importance of understanding MAFLD biology, the functions and molecular modulators of m6A modification, which are suitable contents for the topic of discussion. However, I found the overall organization of the manuscript needs improvement, as well as its clarity and depth.

Thank you so much for the good evaluation to the manuscript.

  1. The title indicates that noncoding RNAs will be the key subject of the review. The authors then started the introduction from circular RNAs (circRNAs), a specific subgroups of RNAs that have both non-coding and coding activities. This could confuse the reader by leaving an impression that circRNAs is the main subject of the review and contradicting the title. According to the title, readers would expect to learn about the involvement of various types of non-coding RNAsin MAFLD. If the authors are only focusing on circRNAs, they should change the title accordingly.

Improved, thank you so much.

  1. The authors tend to discuss the relationship between ncRNA/m6A and MAFLD in a general manner without being specific enough. The authors should explicitly say which circular RNA species and which m6A-modified RNA species impact MAFLD pathology in which direction, through which mechanisms. The authors should avoid statements such as line 223-226 and 260-267. Several good examples are: line 120-123, m6A of CES2 in lipid metabolism; line 142-143, miR-34a-5p in mitochondrial homeostasis (here should introduce the molecular function of miR-34-5p in more detail); line 159-161, circ_0057558 as miR-206 sponge and relevant signaling pathway; line 213-215 miR-183-5p in hepatic triglyceride accumulation, etc. And the authors should organize these pieces of evidences in a logical manner.

True, that may be important. Accordingly, a table relevant has been added in the manuscript. Our point of this paper, however, indeed stands in the “Future perspectives” section. As you can see for the one aspect of topic, we think “microbiota in MAFLD” is the most important argument in this manuscript. Hence, others are largely styled in general for saving the space. Besides, there are excellent reviews elsewhere describing the specific and detailed relationship between ncRNA and liver diseases.

  1. The authors tend to discuss m6A/ncRNA without in the MAFLD context. For example, line 127-138, line 166-178 and line 189-212 all had this issue. It is necessary to introduce the biological background of the subject, such as the role of circRNA and m6A and how they are processed, but to a brief extent. The key is to discuss the involvement of these processes in MAFLD progression, pathology, prognosis etc with specific examples and mechanistic details.

In those paragraphs where you mentioned above, there might not be directly related to the MAFLD content. However, we believe that these paragraphs are needed for understanding the relationship between the technical terms, such as APRO, Tob1, Tob2, BTG1, BTG2, PABP, UTR, miRNA, or ORF, and the MAFLD content. Some mechanistic details have been describing afterwards. Yet, we have furthermore added the discussion of the involvement to MAFLD as much as possible.

  1. The authors tend to confuse between "m6A in MAFLD" versus "the proteins that read/write/erase m6A in MAFLD". The authors spent two sections discussing YTHDF2, an m6A reader, and MAFLD and Diabetes mellitus. Although YTHDF2 is m6A reader, "m6A in MAFLD" and "YTHDF2 in MAFLD" are two distinctive topics. In "m6A in MAFLD", the subject of discussion should be the RNAs with m6A modification, their biological functions and involvement in MAFLD pathology. The authors should change their title and highlight YTHDF2 in their title if this protein is their subject of discussion, and reorganize the manuscript accordingly. 

Very good point! Thank you so much. The title of subsection has been amended, accordingly.

  1. The content of future perspectives is not suitable. Future perspectives should mainly provide the readers with a summary of the future of the research subject discussed in the manuscript, such as challenges and how to address them, opportunities and how to explore them, etc. The authors discusses microbiota in MAFLD, which is only one aspect of topic, and it lacks the depth required for a future perspective section.

We agree to the suggestion of “Future perspectives” in general. In a limited space of manuscript, however, we would like to emphasize the microbiota in MAFLD for the most promising topic in the “Future perspectives” section. Therefore, the last portion about gut microbiota has been improved with additional references.

Comments on the Quality of English Language

The English language is fine, except that the authors tend to use too many "might" "could" in the manuscript, such as: might: line 31, 38, 133 and 134, line 167, line 308, etc; could: line 34, 44, 49, 113, 118, 119 etc. When discussing scientific facts that are supported by previous research data, the authors should avoid being ambiguous or too conservative by using "might" and "could", but be explicit about the facts.

The expression of "might" or "could" in the manuscript has been improved and/or reduced as much as possible.

There are also some expressions that are not commonly used: line 11, considered by; line 11, has been increased lately; line 20, has been appeared to occur; etc.

Improved, thank you.  

Reviewer 3 Report

Comments and Suggestions for Authors

Major comments:

The topic is very interesting and should be addressed, but this review in general is confusing and hard to follow and understand. The sentences are too long and not clear enough. The sections of the document need improved organization, as it is difficult to understand the point and purpose of each paragraph. More and better schemes and tables are needed. The impact of ncRNA and m6A modification on the pathophysiology of MAFLD should be clearer and better explained, with additional details on the studies mentioned. Please pay attention to writting and typing errors (NAFLD instead of MAFLD, font size...).

Minor comments:

The introduction is a bit confusing as you first introduce circRNAs and then ncRNAs in general. 

l40: please explain what NAFLD means.

l48: a reference is missing

Fig 1: this figure is a little confusing because you haven't defined what m6A+ is. The full name of YTHDF2 should be mentioned. The involvement of autophagy leading to MAFLD is not explained either.

l71: this reference does not seem very appropriate.

l116: a reference is missing

l114-l125: it is not clear how m6A modification impacts hepatocyte proliferation and/or differentiation.

l147: is it possible to elaborate a bit more with some examples?

l189-l226: this part is very difficult to follow. You should provide a clearer figure and also a table so that you can properly understand the various interactions and their pathophysiological consequences.

l231: There's no need to repeat the full name of YTHDF2, as it's already been mentioned in l107.

l279: What is the acronym for TME?

l283 and 332: use MAFLD instead of NAFLD

The last part on the impact of the m6A modification on the microbiota is very confusing and should be rewritten.

Author Response

For Reviewer 3

Major comments:

The topic is very interesting and should be addressed, but this review in general is confusing and hard to follow and understand. The sentences are too long and not clear enough. The sections of the document need improved organization, as it is difficult to understand the point and purpose of each paragraph. More and better schemes and tables are needed. The impact of ncRNA and m6A modification on the pathophysiology of MAFLD should be clearer and better explained, with additional details on the studies mentioned. Please pay attention to writting and typing errors (NAFLD instead of MAFLD, font size...).

Typing errors have been amended as much as possible. Thank you so much for the good evaluation to the manuscript.

Minor comments:

The introduction is a bit confusing as you first introduce circRNAs and then ncRNAs in general. 

True, it makes sense. The title of this manuscript has been altered accordingly.

l40: please explain what NAFLD means.

The non-alcoholic fatty liver disease (NAFLD) was introduced in section 1. Afterwards, the term “NAFLD” has been replaced with metabolic disorder-associated fatty liver disease (MAFLD)

l48: a reference is missing

A reference has been added accordingly. Thank you.

Fig 1: this figure is a little confusing because you haven't defined what m6A+ is. The full name of YTHDF2 should be mentioned. The involvement of autophagy leading to MAFLD is not explained either.

Improved, thank you so much.

l71: this reference does not seem very appropriate.

The reference has been replaced with the appropriate one.

l116: a reference is missing

A reference has been added accordingly. Thank you.

l114-l125: it is not clear how m6A modification impacts hepatocyte proliferation and/or differentiation.

According to this suggestion, we have improved the description at that place.

l147: is it possible to elaborate a bit more with some examples?

According to this suggestion, we have enhanced explanation to this sentence.

l189-l226: this part is very difficult to follow. You should provide a clearer figure and also a table so that you can properly understand the various interactions and their pathophysiological consequences.

I am so sorry that the figure and explanation may be difficult to follow. As I had discovered some members of APRO family about 30 years ago, still I think they are curious and interesting molecules with unknown pathophysiological roles. Description in this manuscript is the most understandable one by us. Our intention is that APRO family might be implicated in the cytoplasmic mRNA deadenylation and/or RNAs turnover with circRNAs and ncRNAs.

l231: There's no need to repeat the full name of YTHDF2, as it's already been mentioned in l107.

Corrected, thank you.

l279: What is the acronym for TME?

Accordingly, tumor microenvironment (TME) has been mentioned at the place.

l283 and 332: use MAFLD instead of NAFLD

Corrected, thank you.

The last part on the impact of the m6A modification on the microbiota is very confusing and should be rewritten.

According to the suggestion, the last part about gut microbiota has been extensibely improved with additional references.

Round 2

Reviewer 2 Report

Comments and Suggestions for Authors

The revised manuscript is improved and the authors have provided answers to my questions.